# An Empirical Study of the Contribution of Total Quality Management to Occupational Safety and Health Performance in Saudi Organizations

**DOI:** 10.3390/ijerph20021495

**Published:** 2023-01-13

**Authors:** Mohamed Aichouni, Mabrouk Touahmia, Serhan Alshammari, Mohamed Ahmed Said, Ahmed Baha Eddine Aichouni, Mohsen Almudayries, Hamza Aljohani

**Affiliations:** 1College of Engineering, University of Hàil, Hàil 55255, Saudi Arabia; 2Mechanical Engineering Department, Faculty of Sciences and Technology, University of Coimbra, 3000-531 Coimbra, Portugal; 3Systra Co., Neom Project, Neom 49622, Saudi Arabia

**Keywords:** occupational safety and health, total quality management, empirical study, Saudi organizations

## Abstract

Working to ensure a safe and secure work environment for workers and employees has become an essential component of achieving organizational excellence in modern organizations. Occupational safety and health (OSH) programs help in attracting and retaining the workforce and human competencies, thus improving the operational and organizational performance of these organizations. Total quality management (TQM) is a management philosophy adopted by successful organizations to achieve sustainable business performance. This study aims to assess the level of implementation of total quality management and occupational safety and health in Saudi organizations and investigates the relationship between total quality management practices and occupational safety and health performance. Data were collected from a number of random organizations during the period November to December 2021. Based on a sample of 99 valid responses, empirical results were obtained through descriptive and advanced statistical analysis, indicating that TQM practices and OHS are highly implemented in Saudi organizations. The statistical results also showed that TQM practices have a significant positive impact on OSH performance in the surveyed organizations. The seven TQM fundamental pillars and the five OSH program components can be considered as essential success factors and fundamental pillars for TQM implementation in organizations and for OSH performance improvement.

## 1. Introduction

Recent statistics issued by the International Labor Organization indicate that approximately 7600 people worldwide die daily due to work-related accidents or diseases. About 3,000,000 people are exposed annually to work hazards and accidents in non-working environments. According to David Smith, Chairman of the Committee of Experts on the Development of International Standards for Occupational Safety and Health Management at the International Standards Organization, 2.3 million people die each year because of work-related illnesses or accidents. Accordingly, paying attention to this critical human aspect has become necessary, as it can have a detrimental impact on humans and their subsequent quality of life. Protecting workers from work hazards and supporting their health and safety are fundamental rights guaranteed by laws, legal legislation, and international management systems. Protecting production and operational property and equipment from the risk of accidents and damage contributes to achieving the highest standards of operational excellence for organizations [1].

In a recent article published by the General Organization for Social Security in the Kingdom of Saudi Arabia, the number of workplace injuries during the first half of 2020 was 12,842, with an average of 2140 work accidents per month (or 71 incidents per day); these work-related incidents were scattered throughout the various regions of the Kingdom and spanned across different industrial sectors, such as manufacturing, construction, logistics, and services. According to the report, an estimated 12,001 workers (93.4 percent) who were injured in their workplaces in the private supply chain and logistics sector underwent long-term treatment in hospitals and 683 workers (5.3 percent) recovered without disabilities, while 152 workers (1.2 percent) recovered with permanent disabilities [2]. The enormous numbers of people injured in work accidents are a source of real concern for leadership of organizations, governments, and the whole civil society. This is simply because behind every worker affected by an accident or work illness there are families whose lives may be affected forever due to a work accident or ill-health that happened to that worker [3].

Ensuring and providing a safe and secure work environment for workers in modern organizations has become essential in attracting and stabilizing the workforce and human competencies, thus improving these organizations’ excellent operational and organizational performance. This has a favorable impact on the country’s economic growth and development and contributes to achieving the quality of life targeted by national strategic visions. Given the strategic importance of occupational safety and health for organizations in the era of competitiveness and globalization, governments and specialized organizations have sought to develop strategies, systems, and legislation to contribute effectively and practically to improving the work environment and reducing workers’ injuries. In this context, the Saudi Ministry of Human Resources and Social Development took the initiative to launch the National Occupational Safety and Health Program, with the aim of developing a national system for occupational safety and health for Saudi organizations. The program is based on applying international standards and best practices of the relevant international organizations and developed countries in occupational safety and health. The program mainly contributes to protecting the safety and health of workers, preserving property and the environment, and reducing material losses to organizations [4]. Through the program, organizations can ensure a safe and attractive work environment for workers and contribute to improving safety performance in organizations that suffer from poor safety records.

Total quality management is a management philosophy adopted by successful organizations to achieve sustainable business results. It is mainly based on the implementation of a management system able to deliver consistently high-quality products and services to customers; this is known to contribute to achieving internal and external customers satisfaction, thus increasing the organization’s profitability and market share [5,6,7]. Recent research and world class best practices showed that high quality and consistent products cannot be achieved and sustained unless quality management is implemented in an integrated way with other management systems, especially in areas of occupational safety and health (OSH), environment, and financial and human resource management systems. Most of these management systems have reached an adequate level of development and are well documented in international standards and research published by leading academic and professional organizations worldwide [8,9,10,11,12,13,14,15,16]. However, in the field of operations management research, there has been limited research investigating the relationship and effect of operations management systems and philosophies on the occupational safety and health (OSH) performance in developing countries. There are a very limited number of research studies on the impact of total quality management on occupational safety and health performance in the Arab world in general and in the Saudi context in particular [16,17,18]. To address this research gap, the present study aims are twofold: (a) to perform a bibliometric literature analysis to show that though it is relevant to the Saudi context, this specific research topic is still not covered sufficiently in the literature, and (b) to empirically examine the impact of TQM practices on OSH performance in Saudi organizations. It is believed that the findings from this study will contribute to highlighting the role that TQM practices play in improving OSH performance. These will help Saudi organizations make decisions about how to design, implement, and improve OSH systems and programs that will help them achieve the goals of the Saudi Vision 2030 and its realizations programs [19].

The paper is structured into six sections. Following the introduction in Section 1, the latest literature on total quality management and occupational safety and health performance will be reviewed and presented in Section 2. Section 3 describes the theoretical framework and hypotheses development for the empirical study, followed by the research methodology in Section 4. Section 5 presents the results of the data analysis and hypotheses testing. Finally, we present major conclusions including the implications, limitations, and suggestions for further research.

## 2. Literature Review

This section reviews total quality management (TQM) concepts and research on occupational safety and health (OSH) and how TQM concepts and practices can facilitate and enable occupational safety and health performance in organizations. It starts with a bibliometric review of the two approaches and their implementation worldwide and in the context of Saudi Arabia.

### 2.1. Bibliometric Analysis of TQM and OSH

Bibliometric analysis is the statistical evaluation of published scientific research. The primary aim of bibliometric analysis is to evaluate the importance and impact of the research topic through the analysis of articles that have been published in scientific databases during a specific period. Usually, bibliometric analysis is used to reveal the importance and the relevance of the research, and to assess the attention and interest by researchers, research funding institutions, and practitioners.

The term “bibliometrics” refers to the practice of analyzing bibliographic data from published works of literature through the application of quantitative statistical techniques. Some examples of bibliographic data include the publishing year and the country of origin. In this paper a descriptive analysis of total quality management research fields, focusing on occupational safety and health publications, is presented. As a result, a body of research has been produced, which serves as a foundation for the current empirical study. This part intends to examine the effect of implementing TQM on occupational safety and health performance in organizations. It helps to discover changes, trends, and directions for future research investigations in this specific research area. This method identifies, evaluates, and synthesizes the existing body of published work that researchers, scholars, and practitioners have produced [20]. There is evidence to support the claim that bibliometric reviews have evolved into a “basic scientific activity that would support scientific investigations” [20,21,22].

One of the benefits of using the bibliometrics review approach is that it makes one more aware of the scope of the research and the theoretical foundation for a particular research topic. It is vital to do a bibliometric review of any subject to comprehend the degree of previous research that has been undertaken, as well as to identify the areas that need additional research in the field and flaws that would exist within those areas. In this section, the topic of total quality management within the occupational safety and health area has been implemented using bibliometric analysis [20].

The purpose of this section is to present a bibliometric review of the existing papers in leading journals and specialist journals from 1994 to 2022 to investigate the most common themes in the fields of TQM and OSH. In another aspect, this research aimed to identify the research gap within the context of Saudi business organizations.

The technique suggested for the bibliometric analysis has been outlined in earlier research [21,22] and consists of the following basic steps: (i) study design; (ii) data collection; (iii) data analysis; (iv) data visualization; (v) interpretation. World of Science and Scopus databases were used in order to collect the bibliometric data. Scopus draws its data from the Elsevier Core Collection database, and Web of Science takes it from the Clarivate database. The Scopus and WoS are considered to be primary sources of bibliometric information in many research fields. The collection contains over 500,000 high-quality scientific articles that are peer reviewed and published worldwide in over 240 different fields. The “journal” search for research literature was conducted through academic journals in the fields of total quality management and occupational safety and health management. These top academic journals and specialist journals are published in nine well-known databases, which are as follows: Emerald, American Society for Quality (ASQ), Inderscience, Taylor & Francis, Elsevier, Informs, IEEE Xplore, ProQuest, and John Wiley & Sons. Informs is a database that ProQuest owns.

In October 2022, access was made to the databases and a general search was conducted using the following keywords to narrow the focus: “total quality management” AND “occupational safety and health” OR “TQM” AND “OSH”. After completing an available search by subject, we obtained all of the publications with the keywords in the title and abstract. The analysis took into account documents published between 1994 and 2022 in the databases.

The bibliometric analysis covered a total of 347 documents, which may be divided into two categories: articles (75%) and review papers (10%). Some of the documents have more than one categorization, and one of those classifications is called a conference paper with 10%. The remaining 5% of papers in the databases are editorial content, notes, and book chapters. The journals and books in the Elsevier Core Collection and Clarivate have been indexed in at least one category.

For the aim of this study, we considered some research criteria to make it within the scope of our research area. The research criteria were to consider only articles in the areas of engineering, business management, accounting, and environmental science to cover the occupational safety and health aspects. On the language side, we considered only papers published in English. Finally, we limit our database research to only the keywords “Total Quality Management” and “in Occupational Safety and Health”. Thus, of the more than 50,000 documents retrieved, only 347 have been included in the analysis. For this collection of articles, tags such as author, title, abstract, country, citation record, author affiliation, and author keywords have been saved. Early Access Articles waiting for year assignment were kept in the collection and considered as 2022 since these works may unveil trends. The primary software resource for the bibliometric study was “Bibliometrix”. It is an open-source tool for quantitative research in “Scientometrics” and bibliometrics, programmed in R-tool that provides powerful data analysis and visualization functionalities. In addition, “VoSviewer” was used to identify the author keywords with highest citations [21].

The results of the bibliometric analysis are shown in Figure 1, which shows the annual number of articles published between 1994 and 2022 and the citation number trendline. It can be noted that there is a continuous growth in research related to the contribution of total quality management to the occupational safety and health aspect over time. This growth was apparent from the beginning of the nineties of the last century and the growth continued for nearly twenty years, after which the number continued growing because of the development of the scientific areas of business management. Citations over the years show similar trends. This indicates that the implementation of TQM for OSH performance improvement remains of interest to researchers and professionals worldwide.

These findings demonstrate that though there was a relative drop in the publications’ annual average rate in the last year (2022), there is still interest in the TQM and OSH research though the emergence of new concepts such as Quality 4.0 and Safety 4.0, as a result of the introduction of Industry 4.0 technologies in all organizations management aspects. An average annual growth rate was seen for total quality management and its integration into OSH worldwide, as measured in terms of science and technology output. The annual number of citations has shown a substantial increase, far more than the number of articles. This increase in research interest may be explained by the shift in the management paradigm from normal management to total management, which incorporates TQM, OSH, and environment management systems as drivers and enablers for organizational performance. Even though TQM and OSH research has been going on for 28 years, the field is not showing any signs of slowing down or declining, as measured by the number of scientific publications and research citations. It can also be noted that total quality management has played a fundamental and strong role in everything related to engineering and industrial sciences, emphasizing that it is an integral part of the organizations’ occupational safety and health performance. On the other hand, it is clear that there is a correlation between total quality management and occupational safety and health in general, and this indicates its importance exclusively in processes to achieve organizational goals and sustainable results.

The geographical distribution of the research and work related to the integration of TQM and OSH was addressed. Table 1 examines the prevalence of studies in countries regarding research in TQM and OSH. As shown in Table 1, most scientific publications were issued in Europe (39.24%), the US and Canada (28.07%), and Asia (17.93%). This finding is not surprising, since the common factor among most countries producing research is that they are among global economic leaders. From this figure, it can easily be shown that there is a paucity of research on TQM and OSH in the context of the Middle East and North Africa (MENA) region, with a contribution of (2.19%) of the research. Only a very limited number of studies were recorded in the case of Saudi Arabia, Jordan, and the United Arab Emirates [17,18,23,24]. This finding motivated the authors to perform the present empirical study on the implementation of TQM on OSH performance in the Saudi context.

The last step in the bibliometric analysis was to analyze the author keywords. This procedure aims to define the keywords used in the 347 papers identified in the previous steps. It examines the interaction between the most cited keywords. As shown in Figure 2, which presents the author keywords interactions, two main categories of keywords are identified. The first category contains the major tools and techniques that focus on total quality management. The top five keywords in this category were improvement, quality environment, 5S, Kaizen, and lean manufacturing. The second category is the keywords related to occupational safety and health include safety management system, safety training, total safety management, safety responsibilities, and, finally, risk assessment. From Figure 2, it can be seen that total quality management and occupational safety and health systems are well integrated and interrelated from a research point of view. This would be considered as a good indication that since total quality management has been demonstrated to be an enabler for high quality products and services, it can also ensure a high performance in occupational safety and health programs in organizations [5,6,7,8].

The bibliometric analysis shows the importance and impact of the scientific research and articles published concerning the integrated impact of TQM and OSH. This analysis reveals that the TQM and OSH research areas have received increased interest from researchers, academics, research institutions, and professionals all over the world. This interest is still present in the literature though the new technological advancement brought by Industry 4.0 and the subsequent emergence of new paradigms such as Quality 4.0 and Safety 4.0. In addition, the scarcity of publications dealing with the Saudi context would foster an interesting aspect of the research at both the global and national Saudi levels.

### 2.2. Total Quality Management

Quality is the measure of the capability of all components of an organization to satisfy the stated and implied needs of the customers; it is the measure of how a product or a service will perform satisfactorily in service and is suitable for its intended purpose. The quality concept is not confined to one particular side of the organization, it includes all its operational units, starting with management and ending with resources, production, auditing, stores, and even suppliers and employees. This is achievable only through a commitment to a set of systems divided into units to reach compatibility and harmony between departments and management units [5,6,25]. This is known as total quality management, a modern and holistic management philosophy that originated mainly as a method of work and a means to develop the performance of for-profit and not-for-profit organizations [25,26,27,28,29,30,31,32,33,34,35,36,37,38]. Total quality management (TQM) consists of organization-wide efforts and an integrated system of principles, tools, and best practices to create an environment that supports the organization and continuously improves its capability to deliver products and services that contribute to customer satisfaction [29,30,35,37,38,39].

Various definitions of total quality management have been reported in the literature. Total quality management has been defined as a management approach that focuses on delivering products and services with the highest quality, maximizing customer satisfaction, and meeting national and international regulatory standards. Total quality management is then defined as a business strategy that drives the organization strategy towards achieving customer satisfaction through continuous improvement and full people involvement [6,40,41]. Researchers confirmed that total quality management is about creating a culture of excellence in performance, where management and employees work continuously and diligently to achieve customer expectations and requirements [5,6,7,40,41,42,43,44,45,46,47,48,49,50]. Table 2 summarizes 15 criteria to successfully implement total quality management in manufacturing and service organizations as advocated by Zairi [6].

Quality management scholars and gurus such as Deming, Juran, Crosby, Feigenbaum, Ishikawa, and Zairi have stressed the fact that the customer defines quality. Consequently, total quality management generates customer satisfaction and loyalty, which would lead to building a competitive advantage for the organization. They further stressed that, in conjunction with customer focus, the reduction of the costs of waste and rework are equally important to the organization in achieving its business strategy [6,25,26,40,41,42,43,44,45,46,47,48,49,50]. According to Kiran [50], TQM highlights continuous improvement and a systems perspective to achieve customer satisfaction and long-term organizational success. It involves problem prevention, process improvement, and a team-based approach to problem solving and product improvement, incorporating all departments in the organization. Furthermore, Lim [7] emphasized that TQM is the system of activities aimed to achieve customer delight, empowered employees, higher revenues, lower costs, and improved competitive advantage.

Successful implementation of TQM in world class organizations in services and manufacturing has been reported through (a) scholars’ model (Deming 14 points; Juran 10 steps; and Crosby 14 points) for quality management, (b) business excellence models such as the EFQM, MBNQA, KAQA, Deming Prize, etc., and (c) international standards for quality management (ISO 9001:2015). All these approaches are articulated on a set of fundamental pillars, principles, and practices that constitute the critical success factors for its implementation in organizations [6,7,8]. Depending on the approach for TQM implementation, a multitude of fundamental concepts have been widely agreed upon among scholars and researchers. However, there are common principles mentioned, such as leadership commitment, continuous improvement, process management, customer satisfaction, training and education, teams, and organizational culture [6,7,8,25,40,41,42,43,44,45,46,47,48,49,50,51,52].

To ensure a suitable selection process for the commonly used principles in total quality management implementation, the authors reviewed the TQM fundamental concepts and practices agreed upon in the open literature and from leading professional organizations in quality, such as the American Society for Quality (ASQ, 2022) [28] and the International Standards Organization (ISO) [53]. Table 3 shows the set of the fundamental pillars of TQM that will be used to build the theoretical framework of the present empirical study.

### 2.3. Occupational Safety and Health Performance

In today’s global economy, the organization’s leadership has been increasingly aware of the need for occupational safety and health management systems which aim to create a safe working environment for their workforce. Setting up an effective safety and health management system is crucial in order to reduce problems relating to work accidents and ill conditions in organizations. Safety management systems contribute to the creation of safe working environments and help to reduce incidents, fatality frequencies, and property damage [9,10].

Recent statistics from the International Labor Organization indicated clearly that occupational health and safety is a complex international problem for organizations’ management and the whole of society, and that it must always be a top priority. According to the US Bureau of Labor Statistics, the number of work injuries and illnesses per 100 full-time workers is stable at very high levels, at 2.8 (November 2020 statistics). This is despite the strict application of national or international legal legislation in business environments. The costs associated with these incidents are estimated at billions of dollars annually and represent one of the challenges facing organizations in an era characterized by globalization, intense competition, and technological and social disruptions, which may affect the organization. According to the National Institute for Occupational Safety and Health (NIOSH) in the United States of America, the costs (direct and indirect) of work-related injuries and illnesses were estimated at $170 billion in 2019 [64].

To overcome this situation and avoid these losses, professional experts and organizations have been working hard to develop effective safety systems based on best practices and international standards. Academic research and experts’ work have been devoted to addressing the risks and hazards related to occupational safety and health that organizations face and which negatively affect their performance and even their reputation. Industrial organizations are even required to work hard to achieve the requirements of their customers in products and services, as well as achieving the legal requirements imposed by regulatory and supervisory government authorities regarding the safety of the workforce [14,15,65].

Leading international and professional organizations such as The National Safety Council in the US (NSC) and the ILO provide guidance in the form of fundamental elements that should be addressed in order to establish successful safety and health programs based on the synthesis of research, safety and health expertise, and world class best practices. These elements are recommended to be part of the foundation for an occupational safety and health initiative and program. These elements have been identified by Reese [66] and Aichouni [3] as:

Hazard recognition, evaluation, and control;Workplace design and engineering;Safety performance management;Regulatory compliance management;Occupation health;Information collection;Employee involvementMotivation, behavior, and attitudes;Training and orientation;Organizational communications;Management and control of external exposures;Environmental management;Workplace planning and staffing;Assessments, audits, and evaluations.

According to Reese [64] and Osborne and Zairi [8] it is important for organizations to consider the factors that affect OSH programs and influence their performance; these would be classified under four categories, as shown in Table 4.

Managing occupational safety and health is an integral part of managing the whole business. Organizations need to implement OSH programs for performance and operational excellence through either adopting international standards, such as ISO 45001:2018 for occupational safety and health management systems [65,67], or other models and systems such, as OSH Act or ILO-OSH 2001 Guidelines [66,68,69,70,71,72]. All these OSH management systems or programs are articulated partially or fully on the fundamental elements described in Table 4. The International Labor Organization (ILO) has produced guidelines on the development of occupational safety and health management systems, ILO-OSH 2001, which integrate OSH fundamental elements and total quality management concepts. Such an approach has been widely adopted in research work since the early work of Zairi [5], Weinstein [12,13], Osborne and Zairi [8], Salim et al. [68], and more recently by Aichouni [3]. It is also worth noting that the Saudi National Program for occupational safety and health (OSH) [4] adopts this approach, which is shown in Figure 3.

This international model has been used by leading organizations for achieving excellence in the management of occupational safety and health. It identifies the five interconnected elements which constitute the OSH management system; these elements include: (a) OSH policy and people involvement, (b) organizing, (c) planning and implementation, (d) performance evaluation, (e) improvement. These five elements will be used as constructs for the present research. It can be seen that the framework of this OSH program embraces some aspects of the TQM approach, in the sense that:Policies supportive to human resources development are needed within the OSH program.A systematic approach to OSH management is needed.Continuous improvement of the OSH program is necessary.Supportive organizational culture towards quality and safety and commitment of leadership, with a wide participation of employees at all levels is needed.

## 3. Theoretical Framework and Hypotheses Development

There is strong evidence from the published literature that both total quality management (TQM) and occupational safety and health (OSH) have a great impact on an organization’s performance and its competitive advantage within highly competitive and disruptive economic environments [6].

Since total quality management, with its fundamental concepts, techniques, and management systems, has proven its effectiveness in achieving sustainable results for organizations at all stages of the product lifecycle, it has become necessary to study its integration with other management systems to improve the performance of occupational safety and health programs. Such programs seek to ensure safe and secure work environments for workers, achieving the ultimate goal of zero work accidents.

Research studies and specialized literature that dealt with the occupational safety and health systems indicate a strong relationship between the concepts of occupational safety and health and the fundamental principles of quality management [3,5,7,8,14,15,16,17,18,23,24,41,50]. There is a causal relationship between the two paradigms, meaning that the first causes the second, and the second affects the latter. All these quality management scholars and safety experts agreed on the fact that:Quality in products and services cannot be achieved without achieving safety (No Safety, No Quality). Therefore, safety is an essential requirement to achieve quality.Failure to achieve quality leads to disasters in safety and health (Cost of Poor Quality: Losses in Human Lives, Economic Losses, and Environmental Losses). We all remember the catastrophic Challenger accident on 28 January 1986.

In general, an organization that ignores the safety of its customers and its employees cannot be trusted [7]. In the same way, we cannot trust a brand that does not meet occupational safety and health requirements for its products during the use of those products and during the product lifecycle. Thus, safety is a necessary condition for quality and one of the main attributes that every customer or interested party would look for. Not addressing potential safety issues with an effective management system is like having a deadly disease in the organization system or the delivery service [50]. Additionally, solving potential safety problems during all stages of the product lifecycle, from the product discovery phase to the delivery to the customer through the supply chain process, is a prerequisite for ensuring the quality of operations, products, and after-sales services. Safety and quality go together hand in hand, simply because they are two sides of the same coin. To realize the value of quality in products and services, the organization must prioritize ensuring safety first, achieving efficient and stable processes with differences in output at very few levels, and delivering to customers reliable and value-added product and service quality [5,7].

Based on the literature review of the two management paradigms, TQM and OSH, presented in Section 2, a theoretical framework shown in Figure 4 was developed to investigate the effect of TQM on OSH performance within Saudi organizations. One main research question was formulated on the impact of TQM practices on OSH performance; hence other research questions would consider the level of implementation of both TQM and OSH within Saudi organizations driven by Saudi Vision 2030 momentum. For that purpose, this paper addresses the following research questions:To what extent are TQM and OSH implemented in Saudi organizations?Are there positive relationships among TQM practices and occupational safety and health performance elements?Does TQM implementation have a positive effect on OSH performance in Saudi organizations?

In line with the research gap identified earlier through the literature, and to address these research questions, a set of hypotheses was proposed as follows:

**Hypothesis 1** **(H1):**
*Total quality management is being implemented in Saudi organizations;*


**Hypothesis 2** **(H2):**
*Occupational safety and health management systems are being implemented in Saudi organizations;*


**Hypothesis 3** **(H3):**
*The fundamental concepts of total quality management have relationships among each other while implemented;*


**Hypothesis 4** **(H4):**
*The fundamental elements of occupational safety and health programs have relationships among each other;*


**Hypothesis 5** **(H5):**
*Total quality management positively impacts occupational safety and health performance (OSH).*


The characteristics of organizations are expected to have some effect on TQM and OSH management systems implementation. Organization size, type of business, quality experience, and safety experience can have an influence on TQM and OSH implementation [62,63]. In the present paper no attempt was made to investigate these effects. The authors believe that this analysis would be considered in a further paper.

## 4. Research Methodology

### 4.1. Study Design

The present research aims to assess and measure the degree of implementation of both total quality management (TQM) and occupational safety and health (OSH) in Saudi organizations, and to investigate the impact of implementing TQM on the OSH performance of the organizations. For that purpose, a cross-sectional study of manufacturing, service, and non-profit organizations in Saudi Arabia was performed during the period between November and December 2021.

### 4.2. Data Collection

The theoretical framework developed through the extensive literature review presented in Section 2 was then developed into a survey questionnaire as a data collection instrument. The questionnaire consisted of three main sections; part (1) considers demographic and employment profile of respondents together with the organization characteristics, such as its size, management systems implemented, and its strategic vision with respect to the Saudi Vision 2030; part (2) reflects the implementation of total quality management (TQM) in the respondent’s organization; and part (3) concerns the implementation of an occupational safety and health program (OSH) in the organization. Survey statements required a five-point Likert scale response, in that respondents were required to rate their level of agreement to statements, on a scale of 1 to 5, where 1 = strongly disagreed and 5 = strongly agreed.

Initially, 64 questions were developed to measure the constructs of TQM and OSH. The survey questionnaire was validated through discussion with three academics and professional experts who agreed on 50 final questions. This represented a content validity index (CVI) of 0.781. Hence, the questionnaire was judged to be valid for conducting the investigation. The final version of the survey questionnaire has been circulated to organizations in Saudi Arabia, through electronic media and the LinkedIn social network. The study sample includes a random sample of organizations, either public or private, that operate within the Saudi economic sector and are officially registered with the government (Tadawul Saudi Exchange platform). Out of 500 questionnaires sent in total, 115 questionnaires were received. This makes the response rate approximately 24.75 percent. Such a response rate is considered good enough for conducting research investigations. Following a preliminary analysis of the total number of questionnaires returned, 15 were judged to be incomplete and were dropped from the analysis, bringing the total number of surveys to be analyzed to 99 responses.

### 4.3. Validity Tests

Internal consistency is a commonly used measure in survey analysis. It is an indicator of how well the different items measure the same construct in the survey. Internal consistency is measured by calculating the Cronbach’s alpha statistical factor. Cronbach’s alpha measures internal consistency among a group of items combined to form a single scale. It is a statistic that reflects the homogeneity of the scale. In general, reliability coefficients of 0.70 or more are considered acceptable. In the present study, the Cronbach alpha coefficients of the TQM elements were 0.964, and those of the occupational safety and health (OSH) were 0.976. These values of Cronbach’s coefficients indicate the reliability of the survey scales in yielding valid results for the purpose of the study. Hence, based on the Cronbach’s alpha results, the survey questionnaire was judged acceptable and admissible.

### 4.4. Statistical Analysis Strategy

The survey questionnaire permits us to collect the data from respondents from leadership members and employees in Saudi organizations. The protocol for data analysis is then followed, which consists of the following steps: (a) performing validity tests for the gathered data, (b) discussion of the respondent’s profile and the organization’s characteristics in relation to the Saudi business context and the general economic and social environment occurring in the kingdom. Advanced statistical analysis is then performed for hypotheses testing, using both correlation and regression analyses. Such an approach has been widely described in empirical studies [15,16,24].

## 5. Results

### 5.1. Respondents Profile

The convenience sampling technique was adopted to collect the data, where the target respondents were employees and leadership members of Saudi organizations in different business sectors. Data were collected from small, medium, and large organizations. Initially, a sample size of 500 participants was targeted for assessing the degree of implementation of TQM and OSH. A total of 115 responses were submitted successfully and 99 responses were found complete and appropriate for analysis. The distribution of the respondents according to their gender, age, educational level, position, professional experience, size of organization, and business sector are summarized in Table 5A,B. The organization profile and characteristics related to its experience with quality management and safety, management systems adopted, and strategic planning are summarized in Table 5C.

A careful examination of these tables shows that about 56.6 percent of respondents to the questionnaire were from top and middle management and 43.4 percent were employees. Regarding experience, 60.6 percent of respondents had extensive professional experience in the business, of more than 5 years, and the other 22.22 percent have from 1 to 5 years’ experience in the business. Table 6 shows the cross-correlations of the respondent’s position and their professional experience. It is important to note here that 41.41 percent of the respondents are top and middle management with professional experience of more than five years in the business. These statistics would give some confidence about the results of the present study regarding the implementation of both TQM and OSH in the organizations. Hence, the conclusions would contribute to a better understanding of the actual situation with regards to the implementation of TQM for the improvement of the occupational safety and health programs, which fall within the framework of the Saudi Vision 2030 and the national OSH program. The sampled organizations can be classified as large organizations with more than 500 employees (58.58 percent), medium size organizations (22.22 percent), and small size organizations with less than 100 employees (19.19 percent). The business activities distribution is characterized by the predominance of the industrial sector, with 48.45 percent, and 22.22 percent from the service sector, respectively. Organizations from government represent 17.17 percent, from healthcare, 19.19 percent, from education and training, 9.09 percent, and from the non-for-profit sector, 7.07 percent.

The organizations’ experience in quality and safety management was measured through the survey questionnaire. Of the sampled organizations, 57.6 percent experienced quality management through their quality manager or quality department. A percentage of 77.7 of the organizations have established a safety program and have a safety manager in place. Table 7 also indicates that 64.65 percent of the organizations have a commitment towards both quality management and safety management through assigning both quality and safety managers. It is important to note that only 17.17 percent of the organizations surveyed do not have neither a quality manager nor a safety manager. The surveyed Saudi organizations with recognized management systems, such as ISO 9001:2015 for quality management, represented 53.53 percent of the organizations, followed by OSH MS:ISO 45001:2018, with a percentage of 36.36 percent and EMS ISO 14001:2015 [74], with 22.22 percent. It is believed that these statistics would indicate the high level of adoption of international standards for management systems, as result of the commitment of these organizations towards building the Saudi economy to achieve international competitiveness within the framework of the Saudi Vision 2030. This can be confirmed by the fact that 67.7 percent of the surveyed organizations have a strategic plan in line with the national vision. It can also be noted that 53.54 percent of these organizations with a strategic plan have established a quality management system and have a quality manager and a safety manager in place.

From the above analysis, it can be concluded that quality management practices, such as establishing a QMS and assigning a quality manager, would have a positive impact on driving the organizations towards achieving the national vision 2030. These statistics would clearly indicate that Saudi organizations are committed to both quality management and safety management (hypotheses H_1_ and H_2_). This observation has recently been shown by researchers who empirically investigated the effect of TQM on OSH performance in the MENA region organizations [17,18,23,24] and in the Saudi construction sector [16].

The perception of the respondents of the benefits of TQM and OSH implementation in the organization was measured by the rate of participation in quality and safety applications, and 66.67 percent rate their participation intense participation to moderate participation, while only 11.11 percent have simple to limited participation in quality and safety projects in their workplace. Awareness of the importance of quality and occupational safety and health for Saudi organizations was measured. The results indicate that 42.4 percent of respondents think that TQM and OSH help organizations to comply with regulations, 29.3 percent believe that TQM and OSH help organizations to manage business risks, 16.20 percent perceive that TQM and OSH inspire trust in the business, and 12.1 percent think that TQM and OSH protect the business (Table 8).

### 5.2. Data Analysis and Hypotheses Testing

Prior to testing hypotheses by regression analysis, Pearson correlation analysis was conducted to check correlations among total quality management (TQM) principles. As shown in Table 9, the seven TQM principles are significantly correlated with each other. The correlation coefficients ranged from 0.560 to 0.723 (significant at 0.01 level). It can be seen from Table 9 that all positive coefficients were significant, showing that the correlations between these dimensions were direct. That is, if one element is properly implemented in the organization, it will support the implementation of other elements of TQM. A similar analysis was conducted for the construct of occupational safety and health (OSH) performance, as shown in Table 10. A strong correlation is also found among all the five elements of the OSH program. Both Table 9 and Table 10 show that the components of the TQM approach and OSH program correlate strongly with each other among the surveyed organizations (hypotheses H_3_ and H_4_). This observation would add more strength to our early observation of the wide adoption of TQM practices and OSH elements in Saudi organizations. Such an observation is in perfect agreement with the findings of a recent study presented by Algasseb and Alshmlani [16] who found similar results for Saudi organizations operating in the construction sector. This is believed to be induced by the high momentum generated by the Saudi Vision 2030 in all the economic and business sectors in the kingdom [19].

One of the objectives of the present study is to assess the degree of implementation of total quality management practices and occupational safety and health program elements in the Saudi organizations. The results are summarized for TQM and OSH, respectively, in Table 11 and Table 12. The major observation is that the surveyed organizations show a high level of implementation for TQM fundamental principles, with the highest average of 4.33 for Customer Focus, while the other TQM elements demonstrated a moderate level of implementation. The lowest implementation level was observed for the “evidence-based decision-making” element, with an average of 3.86. The mode values that range from 4 and 5 indicate a perceived high level of implementation of TQM principles. The overall average of the TQM component was 3.95, which indicates a high level of implementation. Similar analysis was undertaken for the OSH program and shown in Table 12. Additionally, it can be seen that there is a high level of adoption of OSH program elements, with an overall average of 3.94 and a mode of 4. This analysis would confirm our early observations derived from the respondents’ profile data, in which it was shown that 64.65 percent of the organizations were committed to total quality management and occupational safety and health through the assignment of quality and safety managers.

Regression analysis was adopted to investigate the relationship between TQM principles and OSH performance and to test hypothesis H_5_. Five multiple regression models were tested in which the independent variables are the seven constructs of TQM practices: Customer Focus (CF), Leadership and Top Management Commitment (LED), People Involvement (PI), Process Approach (PA), Continuous Improvement (CI), Evidence-Based Decision-Making (EBDM), and Relationship Management (RM); and the dependent variables for each model are the OSH program elements (OHS Policy and People Involvement (SPOL), Organizing (SO), Planning and Implementation (SPI), Performance Evaluation (SPE), and Improvement (SI)). The results from the regression analysis are presented in Table 13. It is clear from the analysis that TQM practices have a statistically significant impact on the elements of OSH performance in terms of Policy and People involvement, Organizing, Planning and Implantation, Performance Evaluation, and Improvement at the 5 percent significant level. The R^2^ statistics range from 0.605 for (Safety Planning and Implementation) to 0.704 for (Safety Performance Evaluation). This indicates that, on average, 65 percent of the variance in the dependent variable represented by the OSH program elements are explained by the variation in the independent variables represented by the TQM practices. These findings reflect the fact that the occupational safety and health program investigated here is built on the concepts of total quality management through best practices and benchmarking [8,14,15].

## 6. Conclusions

The present paper presents an empirical study to assess the level of implementation of total quality management and occupational safety and health in Saudi organizations, and to investigate the relationship between TQM and OSH with the purpose of identifying the impact of TQM on occupational safety and health (OSH) performance. Both descriptive and advanced statistical analysis using correlations and regression analysis were used to test the research hypotheses.

The results showed that the surveyed organizations presented high levels of commitment towards total quality management and occupational safety and health implementation. This was measured by the high percentages of organizations that have quality and safety managers, together with a high average score for the seven fundamental pillars of TQM (Customer focus (CF), Leadership and Top Management Commitment (LED), People Involvement (PI), Process Approach (PA), Continuous Improvement (CI), Evidence-Based Decision-Making (EBDM), and Relationship Management (RM)). The five elements of the occupational safety and health (OSH) program (OHS Policy and People Involvement (SPOL), Organizing (SO), Planning and Implementation (SPI), Performance Evaluation (SPE), and Improvement (SI)) scored similar higher averages. The strong correlations between the elements of TQM and the OSH elements indicated a high level of implementation of both TQM philosophy and the OSH performance program in the surveyed Saudi organizations. This is attributed to the high momentum generated by the Saudi Vision 2030 in all economic sectors and the whole society in the kingdom.

The impact of the seven TQM fundamental pillars on the performance of the five elements of the occupational safety and health program was investigated. The statistical results found that total quality management practices have a significant positive impact on occupational safety and health (OSH) performance in the surveyed organizations. The seven TQM fundamental pillars and the five elements of the OSH program can be considered as critical success factors for TQM implementation for OSH performance improvement in organizations.

The authors believe that this study enriches the total quality management (TQM) and occupational safety and health (OSH) literature by proposing and validating measurement instruments for an integrated model based on TQM practices and OSH performance, as well as providing empirical evidence for the relationship between TQM practices and OSH performance. In addition to the results and discussions presented in the paper, management and decision makers in Saudi organizations are provided with insights into the contribution of excellent organizational management models, particularly total quality management, for occupational safety and health performance improvement and reducing work-related injuries and fatalities.

The study contributes to academic research and management practices, yet it is important to raise the limitations noticed by the research group. Characteristics of organizations can have some effect on TQM and OSH management systems implementation. An organization’s size, the type of business activity, quality experience, and safety experience can have an effect on TQM and OSH implementation [15,16,17,18]. In the present paper these effects were not investigated, hence it can be considered one of the limitations. This will be considered in further analysis. The second limitation relates to the measure of TQM practices in the Saudi organizations. The measurements of TQM practices were derived from the literature, which does not directly consider the Saudi context. The authors built their research instrument based on the extensive review of literature published worldwide and customized the survey instrument questions for the Saudi context to be more appropriate for evaluation in a cross-sectional setting, which included various sectors of the economy (industrial, services, NPOs, etc.). This may create some confusion for the respondents from various sectors with different business languages. Additionally, the sample size of the actual study can be considered to be relatively small in terms of generalizing the findings. In addition, common method bias is normally prevalent in empirical studies where data for both independent and dependent variables are obtained from the same respondent in the same measurement context, using the same item context and similar item characteristics. This situation exists in the present study; and it would be necessary to overcome it and check the validity of the results and the findings. Future research would overcome these limitations by refining the survey instrument based on different organizations’ characteristics, collecting the data from a larger sample size, and performing detailed statistical data analyses.

## Figures and Tables

**Figure 1 ijerph-20-01495-f001:**
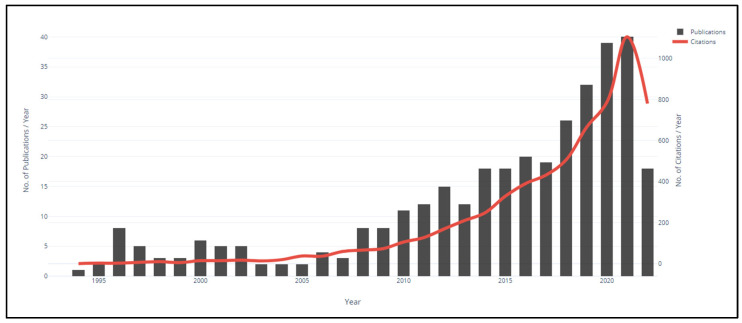
Overall Publication Over Citations (between 1994 and 2022).

**Figure 2 ijerph-20-01495-f002:**
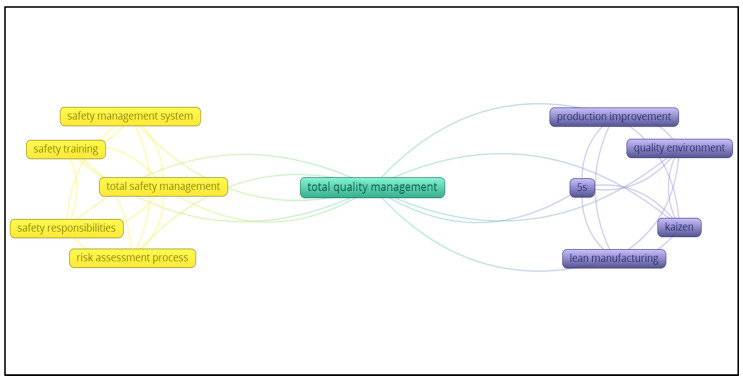
Author Keywords Connections.

**Figure 3 ijerph-20-01495-f003:**
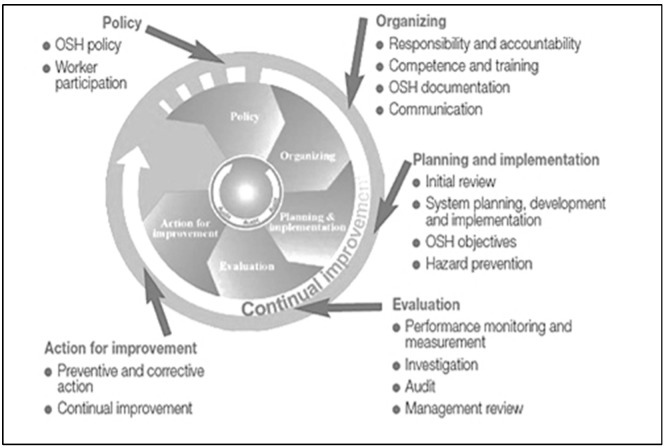
Main Elements of the OSH Program [73].

**Figure 4 ijerph-20-01495-f004:**
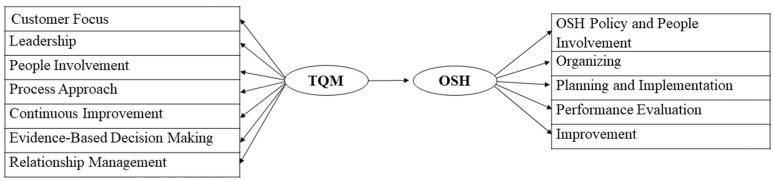
Analytical Research framework.

**Table 1 ijerph-20-01495-t001:** TQM–OHS publications distribution (between 1994 and 2022).

Region	% Publications
North America	28.07
Asia	17.93
Europe	39.24
Middle East	2.19
Africa	4.58
South America	1.39
Australia	6.6
Total	100%

**Table 2 ijerph-20-01495-t002:** TQM implementation as proposed by Zairi [6].

TQM Implementation
TQM is a management philosophy to guide the process of change;
2. Ensures quality as a corporate strategic priority, along with financial and other priorities;
3. Starts at the top;
4. Calls for planning;
5. Requires organization-wide improvement;
6. Calls for everyone to be skilled and knowledgeable;
7. Promotes teamwork;
8. Is about achieving results by process-based approach;
9. Focuses on customer,
10. Recognizes internal customer-supplier relationship;
11. Considers suppliers as part of the organization’s processes;
12. Seeks disciplined approach in continuous improvement efforts;
13. Aims to instill a ‘prevention not inspection” ethic;
14. Emphasizes importance of measurement; and
15. Reduces total cost of meeting customer requirements.

**Table 3 ijerph-20-01495-t003:** TQM principles approved by leading quality organizations and researchers.

TQM Principles	ISO 9001:2015 [53]	ASQ (2022) [28]	Supportive References
Customer Focus	Customer Focus	Customer focused	[6,7,35,37,51,52,54,55]
Employee Involvement	Engagement of People	Total employee involvement	[5,6,7,25,35,37,50,56]
Process Approach	Process Approach	Process Centered	[5,6,7,25,26,50,57,58]
Leadership and Top Management Support	Leadership	Strategic and Systematic Approach	[5,6,7,25,35,50]
Continuous Improvement	Improvement	Continual Improvement	[6,7,35,37,51,52,55]
Evidence-Based Decision-Making	Evidence-Based Decision-Making	Fact-Based Decision-Making	[5,6,7,25,26,35,59,60]
Relationship Management	Relationship Management	Communications	[6,7,35,51,52,55,61]
		Integrated System	[28,56,62,63]

**Table 4 ijerph-20-01495-t004:** Elements of successful occupational safety and health programs (compiled by the authors of [64]).

Element Category	Practical Actions for Implementation
**Management factors**	Management commitment, as reflected by management involvement in aspects of the safety and health program in a formal way and employers’ resources committed to employers’ safety and health program. Management adherence to principles of good management in the utilization of resources (people, machinery, and materials), supervision of employees, and production planning and monitoring. Designated safety and health personnel reporting directly to top management as well as duties and responsibilities from mangers, supervisors, and employees.
**Motivational factors**	Humanistic approach to interacting with employees. High levels of employee/supervisor contact. Efficient production planning.
**Hazard control factors**	Effort to improve workplace safety and health. Continuing development of employees. Clean working environment. Regular, frequent inspections.
**Illness and injury investigations and record-keeping factors**	Investigation of all incidents of illness and injury as well as non-lost-time accidents. Recording and records keeping of all first-aid cases.

**Table 5 ijerph-20-01495-t005:** (**A**) Profiles of the Respondents to the Survey. (**B**) Organizations’ Profile. (**C**) Organizations’ Quality and Safety Practices.

(A)		
Respondents Information	Frequency	Percentage Frequency
**Respondent Gender**		
Male	83	83.8
Female	16	16.2
**Respondent Age Category**		
From 21 to 30	44	44.44
From 31 to 40	40	40.40
From 41 to 50	8	8.08
more than 50	7	7.07
**Educational level**		
Bachelor’s degree	58	58.58
Diploma	18	18.18
PhD and MSc	19	19.19
Others	4	4.04
**Position**		
Top Management	6	6.06
Middle Management	50	50.5
Employee	43	43.4
**Professional Experience**		
>1 year	17	17.2
1–5 years	22	22.22
6–10 years	29	29.29
<10 years	31	31.31
**(B)**		
**Organizations’ Characteristics**	**Frequency**	**Percentage Frequency**
**Size of the organization**		
Less than 50 employees (Small)	19	19.19
From 51 to 500 employees (Medium)	22	22.22
More than 500 employees (Large)	58	58.58
**Organization’s Business Activity**		
Industrial Sector	48	48.45
Services Sector	22	22.22
Civil Society Sector (NGOs)	7	7.07
Governmental Sector	17	17.17
Healthcare Sector	19	19.19
Education and Training Sector	9	9.09
Private Sector	26	26.26
**Geographic Location of Organization**		
Riyadh and Eastern Region	28	28.2
Western Region	37	37.4
Northern Region	11	11.1
Southern Region	23	23.2
**(C)**		
**Respondents Information**	**Frequency**	**Percentage Frequency**
**The organization has a quality manager**		
Yes	57	57.6
No	42	42.5
**The organization has a safety manager**		
Yes	70	70.7
No	29	29.3
**Management system accredited in conformance to international standards**		
ISO 9001 for Quality Management System	53	53.53
ISO 14001:2017 for EMS	22	22.22
ISO 45001:2018 for OHS MS	36	36.36
ISO 26001 for Social Responsibility	9	9.09
ISO 31001 for Risk Management	16	16.16
ISO 22301 for Business Continuity Management System	9	9.09
Other accreditation certificates related to the field of the organization	43	43.43
**The organization has participated in King Abdelaziz Quality Award**		
Yes	33	33.33
No	66	66.67
**The organization has a strategic plan in line with the Saudi Vision 2030**		
Yes	67	67.7
No	32	32.3

**Table 6 ijerph-20-01495-t006:** Distribution of respondents with respect to their position and professional experience (percentages).

	Less Than 1 Year	From 1 to 5 Years	From 5 to 10 Years	More Than 10 Years	All
Top Management	0.00	3.03	1.01	2.02	6.06
Middle Management	5.05	7.07	13.13	25.25	50.51
Employee	12.12	12.12	14.14	5.05	43.43
All	17.17	22.22	28.28	32.32	100.00

**Table 7 ijerph-20-01495-t007:** Percentages of organizations investing in TQM and OSH with respect to assigning quality and safety managers.

	Safety Manager			
Quality Manager		Yes	No	All
	Yes	64.65	9.09	73.74
	No	9.09	17.17	26.26
	All	73.74	26.26	100.00

**Table 8 ijerph-20-01495-t008:** Benefits of adopting TQM and OSH in organizations.

	Frequency	Percentage Frequency
**How do you rate your participation in quality and safety applications in your workplace?**		
I am not interested at all	4	4.0
I want to participate but I do not get the opportunity	7	7.1
I want to participate but I do not have time	11	11.1
Moderate participation	22	22.2
Simple participation and limited in a narrow area of work	11	11.1
Active and intense participation	44	44.4
**The organization has a safety manager**		
Helps protect our business	12	12.1
Helps us comply with regulations	42	42.4
Helps us manage business risk	29	29.3
Inspires trust in our business	16	16.2

**Table 9 ijerph-20-01495-t009:** Correlations among TQM practices.

TQM Fundamental Pillars	CF	LED	PI	PA	CI	EDBM	RM
Customer Focus (CF)	1.00						
Leadership (LED)	0.661 **	1.00					
People Involvement (PI)	0.723 **	0.673 **	1.00				
Process Approach (PA)	0.641 **	0.728 **	0.690 **	1.00 **			
Continuous Improvement (CI)	0.678 **	0.616 **	0.672 **	0.689 **	1.00		
Evidence-Based Decision-Making (EDBM)	0.664 **	0.743 **	0.743 **	0.703 **	0.664 **	1.00	
Relationship Management (RM)	0.560 **	0.616 **	0.678 **	0.680 **	0.631 **	0.781 **	1.00

** Correlation is significant at the 0.01 level (2 tailed).

**Table 10 ijerph-20-01495-t010:** Correlations among OHS Construct.

OHS Performance	SPOL	SO	SPI	SPE	SI
OHS Policy and People Involvement (SPOL)	1.00				
Organizing (SO)	0.852 **	1.00			
Planning and Implementation (SPI)	0.818 **	0.803 **	1.00		
Planning and Implementation (SPE)	0.877 **	0.883 **	0.854 **	1.00	
Improvement (SI)	0.819 **	0.808 **	0.689 **	0.839 **	1.00

** Correlation is significant at the 0.01 level (2 tailed).

**Table 11 ijerph-20-01495-t011:** Summary of TQM statistics.

TQM Fundamental Pillars	Mean	Std Dev	Mode
Customer Focus (CF)	4.33	0.820	5.00
Leadership (LED)	3.88	1.032	4.00
People Involvement (PI)	3.89	1.054	4.00
Process Approach (PA)	3.97	0.963	4.00
Continuous Improvement (CI)	3.89	1.086	4.00
Evidence-Based Decision-Making (EBDM)	3.86	0.989	4.00
Relationship Management (RM)	3.94	1.086	4.00
Overall Statistics for TQM	3.95	1.02	4.00

**Table 12 ijerph-20-01495-t012:** Summary of OSH Performance Statistics.

TQM Fundamental Pillars	Mean	Std Dev	Mode
OHS Policy and People Involvement (SPLO)	3.83	1.05	4.00
Organizing (SO)	3.90	1.04	4.00
Planning and Implementation (SPI)	4.02	0.98	4.00
Performance Evaluation (SPE)	4.05	0.98	5.00
Improvement (SI)	3.92	1.03	4.00
Statistics of OSH Performance	3.94	1.02	4.00

**Table 13 ijerph-20-01495-t013:** Regression analysis of the impact of TQM on Safety Performance.

	OSH Performance				
	OHS Policy and People Involvement (SPOL)	Organizing (SO)	Planning and Implementation (SPI)	Performance Evaluation (SPE)	Improvement (SI)
R	0.800	0.815	0.778	0.839	0.789
R^2^	0.639	0.665	0.605	0.704	0.623
Df.	7	7	7	7	7
Sig.	0.00	0.00	0.00	0.00	0.00
**TQM Fundamental Pillars**	**SPOL**	**SO**	**SPI**	**SPE**	**SI**
Customer Focus (CF)	0.627 **	0.627 **	0.649 **	0.657 **	0.549 **
Leadership (LED)	0.549 **	0.606 **	0.598 **	0.613 **	0.569 **
People Involvement (PI)	0.712 **	0.728 **	0.735 **	0.727 **	0.599 **
Process Approach (PA)	0.636 **	0.681 **	0.640 **	0.692 **	0.606 **
Continuous Improvement (CI)	0.709 **	0.745 **	0.682 **	0.781 **	0.758 **
Evidence-Based Decision-Making (EDBM)	0.625 **	0.657 **	0.650 **	0.697 **	0.648 **
Relationship Management (RM)	0.687 **	0.696 **	0.642 **	0.727 **	0.685 **

** Statistical significance at the 5% level.

## Data Availability

Data can be provided upon request from the corresponding author.

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
