# Peer review of "An Empirical Study of the Contribution of Total Quality Management to Occupational Safety and Health Performance in Saudi Organizations"

_ijerph, 2023, doi:10.3390/ijerph20021495_

Round 1

Reviewer 1 Report

The publication is an article and reading the title it would seem to deal with an empirical case but a large part of the article would seem more like a literature review. 

Two different papers could be created from the article.

line 95: citations are not in order;

line 99: It is not clear if the paper is a systematic literature review or empirical study;

line 120: if it is a systematic literature review, it is necessary to put material and method and then bibliometric analysis.

Make explicit which subdivision the subparagraphs refer to.

line 414: there is no discussion of systematic literature review and paragraph 3 deals with the empirical case.

Author Response

Authors answers to reviewers attached

Reviewer 2 Report

Reviewer’s Comments

12/02/2022

*Issues on the Title

1-The title of this manuscript should be changed “An Empirical Study of the Contribution of Total Quality Management to Occupational Safety and Health Performance—Take Saudi Organizations as An Example.

*Issues in the “Abstract” Section

2-Please remove the texts from line 12 to line 20 before the words “This study” because this description belongs to details instead of summary of the study. Also, for the texts left in this paragraph, the authors should add some analysis results here. For example, the authors mentioned that TQM and OHS are highly implemented in Saudi organizations. Please add a parenthesis between organizations and the symbol “.” and fill it in with the statistical results. All other clauses of the results should be address in the same manner.

*Issues in the “Introductory and Literature Review” Section

3-The authors mentioned that there is few research about the relationship between total quality management and occupational safety and health performance in Saudi Arabia in line 102 on page 2 with the 3 references listed there. However, this is not the whole fact. Please read a systematic review of this topic at https://www.sciedu.ca/journal/index.php/bmr/article/view/15162/9462 for reference. Also, to propose the research hypotheses to address the research gap effectively, the authors should squeeze the content in this section and focus more on Saudi Arabia’s real situation. Otherwise, readers will be overwhelmed with the irrelevant details and can not follow your argument. Please remember you are writing a manuscript of journal paper for publication instead of writing a book chapter. Please remove Figure 1, Figure 2, Figure 3, Figure 4, Figure 6, and Figure 7 in the manuscript and add the content of the questionnaires in the study as appendix.

4-The reviewer suggest the authors propose explicit research questions first and then research hypotheses. Also, note the presentation of the abbreviation for the hypothesis in the parenthesis is incorrect. For example, (H1) should be written as (H1). Furthermore, please replace the rectangles in Figure 5 with two ellipses or circles.

5-Please split “4. Research Methodology and Validity Test” into several parts as follows,

4.1. Study Design― This is a cross-sectional study among manufacturing, service, and non-profit organizations in Saudi Arabi between November 2021 and December 2021.

4.2. Participants and Data Collection. In addition to the details presented in the manuscript, the authors should add the following information: What are the criteria for sample inclusion and sample exclusion? Are there any incentives used for the data acquisition? Did the authors obtain the IRB approval letter before implementing the present study from the affiliation where they work?

4.3 Measures.  

4.3.1 Background characteristics of the participants

The authors should explicitly clarify that the participants were required to report sociodemographic information, including age, ethnicity, relationship status, education level, and whether they were frontline workers or management staff, etc. Note, please clarify if the participants were nested within the organizations in Saudi Arabia. If it is the case, then your data analytical strategy will change accordingly.

4.3.2 Relevant variables

The authors should inform readers how each variable used in the present study was scored and the corresponding metric. For example, how the variable “Customer Focus” was created and what the scoring rubric is.

4.3.3 Please add a subtitle “Statistical Analytical Strategy” and use a or several paragraph(s) to illustrate how the authors would like to analyze the sample. For instance, the 1st step is to blah, blah, blah…. The 2nd step is to analyze XXXX, blah, blah, blah.

*Issues in the “Results and Discussion” Section

6-Please trim the subtitle here as “Results”.

7-Please remove Figure 7 and use a table instead.

8-The Table 13 should be removed. According to the descriptions in the manuscript, the relationship between TQM and OSH should be explored via structural equation modeling instead of regression analysis.

9-Please rewrite the sections “Discussion” and “Conclusion” after re-running the analyses above and correcting all issues that the reviewer pointed out.

Author Response

Authors answers to reviewers attached

Round 2

Reviewer 1 Report

The authors responded to my comments and made my suggested changes.

Author Response

Dear Reviewer

The authors appreciate the reviewer`s constructive comments and thank him for making it clear that his comments have been addressed by the authors, and the manuscript fulfils all the requirements for publication. The reviewer clearly stated that the authors responded to his comments and made all suggested changes. We appreciate it.

Reviewer 2 Report

Reviewer’s Comments

12/27/2022

1-The reviewer described the research questions in the following manner, which would be very straightforward and easily to follow,

In line with these gaps and existing literature, the present study will address the following research questions:

1-Does the proposed factorial structure adjust to the characteristics of the Saudi company population?

2-Does the XXX …….?

If the authors can stand in reader’s position more, think feeling for them, and put yourself into their shoes, in other words, if you were a reader, would you be willing to read a research paper without a clear research question?

2-The authors used Cronbach’s alpha coefficients several times in the manuscript. However, the authors really do NOT know what it is and thus incorrectly present the results. Please test tau-equivalence assumption for sure. Otherwise, it is unreasonable to present the results of Cronbach’s alpha coefficient. When comparing the congeneric and tau-equivalent models, the question is whether constraining the factor loadings in the tau-equivalent model significantly degrades the model fit. If it does, then the assumption of tau-equivalence does not hold. We can use an LR or -difference test to compare the overall fit of the two models. These tests are valid because the congeneric and tau-equivalent models are nested models. Specifically, the tau-equivalent model is nested within the congeneric model. The authors can compute the difference of the chi-squared value and degree of freedom between the congeneric model and the tau-equivalence model, and p-value. If the resultant p-value is less than 0.05, it means that the null hypothesis got rejected.

3- Please compute omega coefficients and replace all alpha’s coefficients with omega coefficients.

4-Please remove Figure 5―It is useless here.

5-Please compute discriminant validity and convergent validity in the section of 4.3.

6-Please check if the common method bias exist. If you really do not know how to do this, please attempt the following methods. Note, you should try all, not one of the three.

1)    Harman’s single-factor analysis

2)    CFA marker variable technique

3)    Controlling for the effects of a single unmeasured latent method factor

Author Response

Dear Reviewer

Thank you for your constructive comments.

Would you please find attached our point by point answers to your comments.

Greetings
